# Unfolded Protein Response Is Activated by Aurora Kinase A in Esophageal Adenocarcinoma

**DOI:** 10.3390/cancers14061401

**Published:** 2022-03-09

**Authors:** Heng Lu, Ahmed Gomaa, Lihong Wang-Bishop, Farah Ballout, Tianling Hu, Oliver McDonald, Mary Kay Washington, Alan S. Livingstone, Timothy C. Wang, Dunfa Peng, Wael El-Rifai, Zheng Chen

**Affiliations:** 1Department of Surgery, Sylvester Comprehensive Cancer Center, Miller School of Medicine, University of Miami, Miami, FL 33136, USA; heng.lu@med.miami.edu (H.L.); ahmed.gomaa@med.miami.edu (A.G.); fxb414@miami.edu (F.B.); txh488@med.miami.edu (T.H.); alivings@med.miami.edu (A.S.L.); welrifai@med.miami.edu (W.E.-R.); 2Department of Chemical and Biomolecular Engineering, Vanderbilt University, Nashville, TN 37235, USA; lihong.bishop@vanderbilt.edu; 3Department of Pathology, Miller School of Medicine, University of Miami, Miami, FL 33136, USA; ogm443@med.miami.edu; 4Department of Pathology, Vanderbilt University Medical Center, Nashville, TN 37235, USA; kay.washington@vanderbilt.edu; 5Division of Digestive and Liver Diseases, Department of Medicine, Columbia University Medical Center, New York, NY 10032, USA; tcw21@cumc.columbia.edu

**Keywords:** esophageal adenocarcinoma, AURKA, ER stress, drug resistance

## Abstract

**Simple Summary:**

Esophageal cancer is the 6th most common cause of cancer-related deaths in 2018 worldwide, with a 5-year survival rate of around 20%. This study reported that esophageal adenocarcinoma cancer cells take advantage of unfolded protein response to survival. Our data using both in vitro cancer cell models and in vivo mice models discovered, for the first time, that Aurora kinase A hijacks pro-survival unfolded protein response in esophageal adenocarcinoma, promoting the survival of cancer cells under reflux-mediated stress conditions.

**Abstract:**

Unfolded protein response (UPR) protects malignant cells from endoplasmic reticulum stress-induced apoptosis. We report that Aurora kinase A (AURKA) promotes cancer cell survival by activating UPR in esophageal adenocarcinoma (EAC). A strong positive correlation between AURKA and binding immunoglobulin protein (BIP) mRNA expression levels was found in EACs. The in vitro assays indicated that AURKA promoted IRE1α protein phosphorylation, activating prosurvival UPR in FLO-1 and OE33 cells. The use of acidic bile salts to mimic reflux conditions in patients induced high AURKA and IRE1α levels. This induction was abrogated by AURKA knockdown in EAC cells. AURKA and p-IRE1α protein colocalization was observed in neoplastic gastroesophageal lesions of the L2-IL1b mouse model of Barrett’s esophageal neoplasia. The combined treatment using AURKA inhibitor and tunicamycin synergistically induced cancer cell death. The use of alisertib for AURKA inhibition in the EAC xenograft model led to a decrease in IRE1α phosphorylation with a significant reduction in tumor growth. These results indicate that AURKA activates UPR, promoting cancer cell survival during ER stress in EAC. Targeting AURKA can significantly reverse prosurvival UPR signaling mechanisms and decrease cancer cell survival, providing a promising approach for the treatment of EAC patients.

## 1. Introduction

Esophageal cancer is the 7th most commonly occurring cancer in men and the 13th most commonly occurring cancer in women globally [1]. It is the 6th most common cause of cancer-related deaths, with over 500,000 new cases in 2018 worldwide [1]. There are two main types of esophageal cancer: squamous cell carcinoma, which occurs in the upper part of the esophagus, and adenocarcinoma, which develops at the junction of the esophagus and stomach [2]. Most esophageal cancers in the United States are adenocarcinomas, while squamous cell carcinoma is globally common [3]. The 5-year survival rate is about 20% for all patients with all esophageal cancer stages, making esophageal cancer one with the lowest 5-year relative survival rate from 2010 to 2016 [4]. As compared with the overall 5-year survival rate for esophageal cancer, the 5-year survival rate of esophageal adenocarcinoma is approximately 17–19% [5].

Gastroesophageal reflux disease (GERD) is a long-term condition where acidic bile salts abnormally reflux into the lower esophagus, causing irritation and chronic inflammation [6]. GERD is considered the leading risk factor for developing a metaplastic glandular condition known as Barrett’s esophagus (BE) with its progression to EAC [7]. Recent studies have demonstrated that exposure of BE and EAC cells to acidic bile salts (ABSs), the mimic of GERD, induces high levels of oxidative stress and DNA damage [8].

Aurora kinase A (AURKA), a member of mitotic kinases, is localized to centrosomes and the mitotic spindle during mitosis. It favors the G2/M transition by promoting centrosome maturation and mitotic spindle assembly [9]. Several studies have indicated that AURKA is overexpressed and amplified in several malignancies [10,11]. There are accumulating lines of evidence supporting the nonmitotic function of high levels of AURKA in cancer cells. The literature has shown that overexpression of AURKA mediates several protumorigenic functions in addition to mitosis [12,13]. AURKA activates multiple oncogenic signaling pathways, including STAT3, NF-kB, and β-catenin, while suppressing critical tumor suppressor functions of p53 and TAp73 in cancer cells, promoting cancer cell survival and drug resistance [11,14,15]. These findings suggest AURKA as a promising therapeutic target in EAC.

Tumor cells can grow under conditions of oncogenic stress caused by DNA damage, hypoxia, and metabolic and oxidative stress. These factors lead to the accumulation of unfolded or misfolded proteins in the endoplasmic reticulum (ER), known as ER stress [16]. Unfolded protein response (UPR) balances the ER folding environment under ER stress [17]. Detecting and resolving ER stress requires three major ER-spanning transmembrane proteins, inositol-requiring enzyme 1a (IRE1a) (encoded by ERN1), PKR-like ER kinase (PERK) (encoded by EIF2AK3), and activating transcription factor 6a (ATF6a) (encoded by ATF6). Each is bound by the chaperone protein BIP, which locks them in inactive states [18]. If the intraluminal misfolded proteins’ level exceeds ER’s folding capacity, BIP dissociates from IRE1a, PERK, and ATF6a. These sensors subsequently drive downstream signaling pathways to correct ER stress [19]. IRE1α phosphorylation and activation is a crucial mechanism promoting cancer cell survival through UPR. Sustained UPR activation provides malignant cells with greater tumorigenic, metastatic, and drug-resistant capacity [17]. Recent studies have uncovered that UPR further inhibits the development of protective anticancer immunity [20]. Targeting prosurvival UPR during ER stress is a novel therapeutic strategy for several malignancies, including colorectal cancer, lung cancer, and leukemia [21,22]. In this study, we report, for the first time, that AURKA hijacks prosurvival UPR in EAC, promoting the survival of cancer cells under reflux-mediated stress conditions.

## 2. Materials and Methods

### 2.1. Cell Culture and Reagents

Two human cell lines originated from Barrett’s esophagus, BART (kindly provided by Rhonda Souza) and CPB (ATCC, Manassas, VA, USA), were cultured with DMEM/F12 (Gibco, New York, NY, USA), supplemented with 5% fetal bovine serum (FBS, Gibco), 1% penicillin/streptomycin (GIBCO), 0.4 μg/mL hydrocortisone (Sigma-Aldrich, Saint Louis, MO, USA), 20 mg/L adenine hydrochloride hydrate (Sigma-Aldrich), 140 μg/mL bovine pituitary extract (Thermo Fisher Scientific, Waltham, MA, USA), insulin–transferrin–sodium selenite media supplement (Sigma-Aldrich), and 20 ng/mL recombinant epidermal growth factor (Sigma-Aldrich). The esophageal adenocarcinoma cell lines FLO-1 (ATCC), OE33 (kindly provided by Dr. David Beer), SK-GT4 (kindly supplied by Dr. Xiaochun Xu at MD Anderson), and ESO26 (Sigma-Aldrich) were cultured in DMEM or RPMI 1640 medium (GIBCO) supplemented with 10% FBS and 1% penicillin/streptomycin. All cell lines were grown at 37 °C in 5% CO_2_. Cell lines were authenticated by Genetica DNA Laboratories using short tandem repeat profiling (Genetica DNA Laboratories, Burlington, NC, USA). Cells used were from stocks immediately after authentication and cultured less than 6 months. Mycoplasma was tested periodically using the qRT-PCR method (SouthernBiotech, Birmingham, AL, USA). Phospho-IRE1α (Ser724) was obtained from Novus Biologicals. AURKA, IRE1α, cIAP2, PARP, cleaved PARP, BCL-2, BIP, XBP1s, phospho-PERK (Thr980), and phosphor-eIF2α (Ser51) BAX antibodies were purchased from Cell Signaling Technology (Beverly, MA, USA). The β-actin antibody was obtained from Sigma-Aldrich. Mouse or rabbit secondary antibodies were purchased from Promega (Madison, WI, USA).

### 2.2. TCGA and GEO Datasets Analysis

TCGA data for AURKA and BIP mRNA expression in EAC tissue samples were analyzed using the online tool UCSC Xena at https://xena.ucsc.edu/, accessed on 1 October 2020 [23]. The mRNA expression data of the esophageal adenocarcinoma (EAC) cohort were obtained from the Cancer Genome Atlas (TCGA) (https://portal.gdc.cancer.gov/, accessed on 1 February 2021) and Gene Expression Omnibus (GEO) (https://www.ncbi.nlm.nih.gov/gds, accessed on 1 May 2021) database. Samples diagnosed with EAC were enrolled for further analysis. The number of EAC samples included in each cohort was as follows: TCGA (*n* = 78), GSE26886 (*n* = 21), GSE37201 (*n* = 22), GSE37200 (*n* = 15), and GSE74553 (*n* = 52). We used R language (R 3.6.1; https://www.r-project.org/, accessed on 1 May 2021) for genetic annotation. When a gene contained multiple probes, these probes’ mean was used for the gene expression value. Differential expression gene analysis was performed with the limma package. To analyze AURKA-related signaling pathways in EAC, we divided the samples of each data set into two groups. High-expression (top 25% AURKA expression samples) and low-expression (lowest 25% AURKA expression samples) groups were analyzed using mean ± SD of the AURKA mRNA value as the cutoff point. The ClusterProfiler v3.12.0 package (https://guangchuangyu.github.io/software/clusterProfiler, accessed on 1 May 2021) [24] was performed for GSEA analysis using DEG comparing AURKA high- and low-expression groups. All hallmark gene sets were obtained from MSigDB (https://www.gsea-msigdb.org/, accessed on 1 May 2021).

### 2.3. RNA Sequencing and Gene Set Enrichment Analysis

A total of 18 RNA samples were extracted from FLO-1, OE33, and SK-GT4 control (triplicates) or alisertib 400 nM 48 h treated (triplicates) cells. RNA seq was performed on a total of 1 µg RNA from each sample as described before [25]. Gene set enrichment analysis (GSEA) [26] was performed based on the RNA sequencing data obtained from all three EAC cell lines. The gene set used for UPR pathway analysis was Gene Set: HALLMARK_UNFOLDED_PROTEIN_RESPONSE (https://www.gsea-msigdb.org/gsea/msigdb/cards/HALLMARK_UNFOLDED_PROTEIN_RESPONSE, accessed on 1 May 2021), which contains 22 genes upregulated during unfolded protein response.

### 2.4. qRT-PCR and Human Esophageal Tissue Samples

Deidentified human tissue samples from 32 esophageal cancer and 32 normal esophageal tissue samples were collected from the National Cancer Institute Cooperative Human Tissue Network (CHTN) and the pathology archives at Vanderbilt University Medical Center (Nashville, TN, USA). All tissue samples were collected, coded, and deidentified in accordance with the Institutional Review Board-approved protocols. The histology and age information are included in Appendix A. Total RNA was purified using the miRNeasy mini kit (Qiagen, Redwood City, CA, USA). miRNA cDNA was reverse-transcripted as described before [27]. Quantitative real-time PCR (qRT-PCR) was performed using a Bio-Rad CFX Connect Real-time System with the threshold cycle number determined by Bio-Rad CFX manager software version 3.0. Primers that detect human genes were ordered from Integrated DNA Technologies (Coralville, IA, USA). The genes and sequences of qRT-PCR primers are given in Appendix A. Results of target genes were normalized to human HPRT1.

### 2.5. Cell Viability ATP-Glo and Clonogenic Cell Survival Assay

Control siRNA or AURKA siRNA transfected FLO-1, and OE33 cells were seeded at 1000 cells per well in 96-well plates and treated with tunicamycin (range: 0.3–300 ng/mL) or AURKA inhibitor AK-01 (LY3295668) (range: 0.20–200 nM) or AURKA inhibitor alisertib (rang: 7.8–500 nM) or IRE1α inhibitor 3′6′-DMAD (range: 15.6–1000 nM) or PBS (control) for 5 days. Cell viability was measured using the CellTiter-Glo Cell Viability Assay (Promega, Madison, WI, USA). Changes in absorbance were recorded in a FluolarStar luminescence microplate reader (BMG Labtech, Ortenberg, Germany). FLO-1 or OE33 cells were seeded 500 cells/well in 6-well plates treated with tunicamycin (range: 0.0–400 ng/mL) or alisertib (range: 0–250 nM) or a combination for 48 h. Following treatments, cells were washed with PBS following incubation in a drug-free DMEM cell culture medium for 10 days. Subsequently, the media were removed, and cells were fixed with 4% paraformaldehyde solution for 10 min at room temperature. The cells were then gently washed with PBS and stained overnight with crystal violet (0.05% crystal violet in 50% methanol). Following overnight staining, the excess dye was gently washed off with PBS. The plates were photographed. Colony formation and cell survival were evaluated by quantifying the dye signal in each well with ImageJ image analysis software (https://imagej.nih.gov/ij/, accessed on 1 May 2021).

### 2.6. Western Blotting

Cells were collected after 5% trypsinization of the culture plate, followed by centrifuging at 12,000 rpm at 4 °C for 10 min. Cell pellets were resuspended in RIPA buffer containing protease inhibitor cocktail and phosphatase inhibitor sodium orthovanadate (Santa Cruz Biotechnology Inc., Dallas, TX, USA) on ice for 30 min with vortex every 10 min. Protein concentrations were measured using a Bio-Rad Protein Assay (Bio-Rad Laboratories, Hercules, CA, USA). An amount of 20 µg proteins from each sample were subjected to SDS-PAGE and transferred onto nitrocellulose membranes (PerkinElmer, Waltham, MA, USA). Membranes were blocked with 5% bovine serum albumin (BSA, Sigma-Aldrich). To detect target proteins, membranes were probed with specific primary antibodies overnight. The next day, membranes were washed for 10 min with TBS-T 3 times, followed by incubation with anti-rabbit or anti-mouse secondary antibodies. Protein bands were detected using chemiluminescence reagents (Millipore, Billerica, MA, USA).

### 2.7. Flow Cytometry Analysis of Cell Apoptosis/Death

Flow cytometry analysis of Annexin V and PI was performed using a FITC Annexin V apoptosis detection kit (BD Biosciences, San Jose, CA, USA) to quantitate ABS-induced apoptosis. EAC cells were treated with 200 nM alisertib or control. After 24 h, cells were treated with pH 4.0, 200 µM ABS, for 20 min, then recovered in regular medium for 3 h. According to the manufacturer’s instruction, cells were then collected for FITC Annexin V and PI staining and subjected to flow cytometry analysis at the Flow Cytometry Shared Resource at the Sylvester Comprehensive Cancer Center.

### 2.8. Immunofluorescence

All tissue slides were prepared by the pathology core at the Sylvester Comprehensive Cancer Center. Immunofluorescence was utilized following standard protocols [28]. Paraffin-embedded slides were deparaffinized after 3 × incubation (3 min) in Histo-Clear (National Diagnostics, Atlanta, GA, USA), 2 × (2 min) in 100% ethanol, 2 min in 95% ethanol, 2 min in 70% ethanol, 2 min in 50% ethanol, and 2 min in double deionized water (i.e., ddH_2_O). Additionally, slides were incubated in TE buffer, pH = 8.0, for 10 min at 100 °C for antigen unmasking. Slides were cooled to room temperature after antigen unmasking, and 5% bovine serum albumin was used to block slides for 1 h. Slides were incubated with p-IRE1α (S724, Novus Biologicals, Littleton, CO), AURKA (Cell signaling Technology, Danvers, MA, USA), or Ki-67 antibody (Cell Signaling Technology, Danvers, MA, USA) 1:250 overnight at 4 °C. After washing with PBS 3 times, the slides were incubated with 1:500 goat anti-rabbit Alexa Fluor 488 and goat anti-mouse Alexa Fluor 568 (Thermo Fisher Scientific, Weston, FL, USA) secondary antibodies for 2 h in the dark. After this procedure, the slides were covered with a mounting medium with 4′,6-diamidino-2-phenylindole (Vector Laboratories, Burlingame, CA, USA). Images were taken using an FV-1000 confocal microscope (Olympus America, Miami, FL, USA). ImageJ software was used for Ki-67 data quantification.

### 2.9. Animal Models

For tumor xenograft and alisertib treatment, FLO-1 and OE33 cells (4 × 10^6^) were suspended in a 200 µL DMEM–Matrigel mixture (50% DMEM, 50% Matrigel). Cells were injected into the flank regions of female athymic nude-Foxn1 nu/nu mice (Harlan Laboratories Inc., Indianapolis, IN, USA). Alisertib treatment was started with daily alisertib (30 mg/kg, orally) for 3 weeks when the tumor reached 150–200 mm^3^. Tumor size was calculated as the formula: tumor volume = (length × width^2^)/2. Each treatment group included at least 10 tumor xenografts. The tumor regression results were partially reported earlier [29].

The pL2-IL1β transgenic mice were a kind gift from Dr. Timothy Wang (Columbia University), a model of chronic esophageal inflammation that develops BE and EAC, as previously described [30]. The mice received drinking water containing 0.3% deoxycholic acid (DCA) at neutral pH for 7 months. Then the mice were sacrificed and subjected to histological analysis of the squamocolumnar junctions at the gastroesophageal junctions. All animal work was approved by the Institutional Animal Care and Use Committee.

## 3. Results

### 3.1. AURKA Overexpression Correlates with UPR Activation in EAC

We and others have shown that AURKA plays vital biological functions to promote cancer cell survival [9,14]. To further identify novel roles of AURKA in EAC, AURKA mRNA expression levels at TCGA and four GEO databases containing 188 human esophageal adenocarcinomas tissue samples were analyzed. We performed signal pathway enrichment analysis using the upper quartile of AURKA^high^ expressing compared with the lower quartile of AURKA^low^ expressing EAC samples. The results demonstrated that the unfolded protein response (UPR) pathway was highly activated in AURKA^high^ EAC samples compared with AURKA^low^ tissues (Figure 1A, left and middle panels, *p* < 0.01). To further test the correlation between AURKA and UPR in EAC, next-generation sequencing (NGS) analysis was performed between AURKA inhibitor (alisertib) treatment groups and control groups in FLO-1, OE33, and SK-GT4 cells. Gene set enrichment analysis using NGS data from the three cell lines demonstrated that unfolded protein response gene set expression was significantly downregulated in alisertib-treated EAC cells compared with control EAC cells (Figure 1A, right panel, *p* < 0.001). UPR is known to be activated under ER stress in both normal and cancer cells [31,32]. Since few reports are published on the relationship of AURKA and ER stress, we further looked at our esophageal tissue samples and the TCGA database for BIP mRNA, a marker of ER stress [33] AURKA mRNA expression in EAC. Using RT-PCR in 30 normal esophageal samples and 32 EAC tissues, we detected a significant overexpression of both AURKA and BIP mRNA in EAC samples compared with the normal esophagus (NE) (Figure 1B, *p* < 0.01). The results showed a strong positive correlation between AURKA and BIP mRNA in esophageal tissue samples (Figure 1C, *p* < 0.0001). TCGA data analysis also demonstrated that AURKA mRNA was significantly overexpressed in EAC samples compared with normal esophageal tissues as expected (Figure 1D, left panel, *p* < 0.001). In the meantime, BIP mRNA was significantly more expressed in EAC than normal esophageal samples (Figure 1D, right panel, *p* < 0.01). The analysis demonstrated a significant positive correlation of AURKA mRNA and BIP mRNA expression in esophageal tissue samples (Figure 1E, *p* < 0.0001, 11 normal esophagus samples, 79 esophageal adenocarcinoma samples). These findings suggest that UPR activation is positively correlated with AURKA expression in EAC samples. Based on these findings, we hypothesized that AURKA plays an essential role in activating UPR in EAC.

### 3.2. IRE1α Promotes EAC Cell Survival through AURKA

To test our hypothesis and further investigate the role of AURKA in the UPR pathway, we examined the protein expression levels of AURKA and critical UPR signaling proteins in three normal esophageal tissues (NE1, NE2, and NE3), Barrett’s esophagus (BART and CPB), and esophageal adenocarcinoma cell lines (FLO-1 OE33, SK-GT4, and ESO26). Western blot results demonstrated that AURKA, p-IRE1α (S724) IRE1α, BIP, XBP1s, p-PERK (T982) PERK, and p-eIF2α(S51) protein expression levels were remarkably higher in Barrett’s esophagus and EAC cells compared with normal esophageal tissues (Figure 2A). IRE1α, PERK, and ATF6 are three well-known sensors that activate UPR, controlling the balance between cell survival and apoptosis during ER stress in cancer [33,34]. To test which sensors are vital in EAC cell models, we knocked down these proteins respectively in EAC cell lines using siRNA. Western blot data clearly showed that knockdown of PERK or IRE1α was successful in FLO-1 and OE33 cell lines (Figure 2B). Interestingly, in both cell lines, IRE1α knockdown remarkably induced cleaved PARP expression, while the PERK or ATF6 knockdown had little effect on cleaved PARP induction (Figure 2B,C). These results suggest that IRE1α is essential for EAC cell survival. To test whether AURKA plays a vital role in regulating IRE1α, we used AURKA siRNA knockdown in FLO-1 or OE33 cells. Western blot data demonstrated that AURKA knockdown decreased IRE1a phosphorylation and protein levels in both cell lines (Figure 2D). To further test our hypothesis that AURKA promotes EAC cell survival during ER stress by inducing and activating prosurvival IRE1α expression, we used tunicamycin (TM) [35] to generate ER stress with or without AURKA siRNA knockdown in FLO-1 or OE33 cells. Western blot data indicated TM treatment-induced BIP protein expression in both cell lines, as expected. In the meantime, AURKA knockdown decreased the IRE1α phosphorylation, expression, and protein expression level of cIAP2, a known downstream prosurvival target of IRE1α activation [36] (Figure 2E). Interestingly, AURKA knockdown induced higher levels of cleaved PARP with TM-treated cells than control siRNA (Figure 2E). These data strongly support our hypothesis that AURKA promotes IRE1α activation and cell survival in EAC cells.

### 3.3. IRE1α Activation Is Dependent on AURKA in Response to Acidic Bile Salt Reflux Conditions in EAC Cells

Acidic bile salts (ABSs) in gastroesophageal reflux disease (GERD) cause esophageal irritation and inflammation, the leading risk factor for EAC. ABSs induce DNA damage and oxidative stress in EAC cells. Recent studies have shown that cells under oncogenic stress, such as DNA damage and oxidative stress, develop ER stress, which activates UPR [37,38]. In this study, we found that AURKA induced IRE1α activation in EAC cells. Therefore, we investigated whether ABSs induce UPR in EAC and whether this induction is dependent on AURKA. FLO-1, OE33, and SK-GT4 cells were treated with pH 4.0 200 µM ABSs for 20 min to mimic a typical reflux episode in patients with GERD. The culture media were replaced with fresh regular media without ABSs after the 20 min treatment. Cells were harvested right after 20′ ABS treatment (0 h) or with 1 h (1 h) or 3 h (3 h) recovery time points after the 20′ ABS treatment. Western blot results demonstrated that ABS treatment induced AURKA protein expression (Figure 3A,B and Appendix A, left panels). Interestingly, there was a remarkable induction of IRE1α phosphorylation (S724) in all three EAC cell lines at different time points (0 to 3 h) after ABS treatment (Figure 3A,B and Appendix A, left panels). The AURKA knockdown in all three cell lines abrogated ABS-induced IRE1α phosphorylation at S724 (Figure 3A,B and Appendix A, left panels). To further confirm the activation of IRE1α in our settings, we examined the ratio of spliced xBP1 mRNA and total xBP1 mRNA expression levels in the three cell lines as a measure of IRE1α downstream activity. RT-PCR data demonstrated that ABS treatment significantly induced the ratio of spliced xBP1 mRNA and total xBP1 mRNA in FLO-1, OE33, and SK-GT4 cells (Figure 3A,B and Appendix A, right panels, *p* < 0.001). We treated FLO-1 or OE33 cells with two different AURKA inhibitors, alisertib or AK-01, with or without ABS to further validate our findings. Treatment with alisertib or AK-01 abrogated ABS-induced IRE1α levels in FLO-1 and OE33 cells (Figure 3C,D and Appendix A). For the first time, these data demonstrated that AURKA mediated ABS-induced activation of IRE1α in EAC cells. These findings suggest that EAC cells highjack the AURKA–IREα–xBP1 axis to promote survival and overcome reflux-induced stress.

### 3.4. AURKA Protects EAC Cells from ABS-Induced Apoptosis

To investigate the prosurvival role of AURKA in EAC cells during ABS treatment, we used Annexin V/PI staining assay with or without ABS treatment (20′ treatment + 3 h recovery), AURKA inhibitor alisertib (400 nM overnight pretreatment), or combinations. The results indicated that ABS induced cancer cell death as measured by Annexin V staining, PI staining, and double staining (Figure 4A,B and Appendix A, *p* < 0.01). In the meantime, cell death was significantly higher with AURKA siRNA knockdown with or without ABS treatment compared with control siRNA (Figure 4A,B and Appendix A, *p* < 0.01). Using Annexin V/PI staining in SK-GT4, ABS treatment induced significantly more cell death in AURKA siRNA knockdown than control SK-GT4 cells (Figure 4C).

### 3.5. AURKA Binds to IRE1α in EAC Cells

To investigate the mechanisms by which AURKA phosphorylates IRE1α, we performed proximity ligation assay (PLA) for AURKA and IRE1α proteins in both FLO-1 and OE33 cells. Our data indicated that AURKA was closely localized with the IRE1α protein in both cell lines (Figure 5A and Appendix A). Furthermore, ABS treatment significantly increased PLA signals. In contrast, the alisertib treatment diminished such induction in both FLO-1 and OE33 cells, consistent with our findings that ABS-induced IRE1α phosphorylation is dependent on AURKA in EAC cells (Figure 5A,B and Appendix A, *p* < 0,05). To further validate the binding of AURKA with IRE1α, we performed immunoprecipitation (IP) in FLO-1 cells with or without ABS or alisertib treatment and OE33 cells with or without ABS or AURKA siRNA knockdown. Our Western blot data demonstrated that ABS induced IRE1α phosphorylation in both cell lines. The induction of IRE1α phosphorylation was abrogated by alisertib treatment or AURKA siRNA knockdown, as expected (Figure 5C and Appendix A). In the meantime, ABS remarkably promoted the binding of AURKA and IRE1α, detected by Western blot in IRE1α IP samples. Alisertib treatment or AURKA knockdown diminished the induction of binding between AURKA and IRE1α (Figure 5D and Appendix A). Our data also indicated no change of binding between BIP and IRE1α with ABS or alisertib treatment in FLO-1 cells (Figure 5D). We also performed in vitro kinase assay to investigate whether AURKA phosphorylates IRE1α directly. However, we cannot conclude due to the autophosphorylation of IRE1α in the experimental condition. Autophosphorylation of IRE1α is well [39,40]. These results suggest that AURKA binds to IRE1α to promote its autophosphorylation in EAC cells. This binding is strongly induced with ABS treatment.

### 3.6. AURKA Promotes Survival under Tunicamycin-Induced ER Stress in EAC Cells

As the results indicated that AURKA induced IRE1α phosphorylation, we investigated the potential prosurvival role of AURKA under cytotoxic ER stress in EAC cells. FLO-1 or OE33 cells were treated with the ER stress inducer tunicamycin (TM, 250 or 500 ng/mL, 24 h) with or without AURKA siRNA knockdown. ATP-Glo results demonstrated that TM alone significantly decreased cell viability on day 3 in both cell lines (Figure 6A and Appendix A, *p* < 0.01). The combination of AURKA knockdown and TM significantly reduced cell viability more than AURKA siRNA alone, TM alone, or the control group (Figure 6A and Appendix A). Cell viability was also determined with different concentrations of TM, AURKA inhibitor AK-01, or a combination. Our ATP-Glo data indicated that combination treatment decreased TM IC50s in FLO-1 (from 74.2 to 23.99 ng/mL) and OE33 cells (from 38.69 to 23.65 ng/mL) (Figure 6B and Appendix A). The IC50s of AK-01 were decreased in combination groups compared with single treatment (FLO-1: from 25.96 to 15.99 nM; OE33: from 392.24 to 63.14 nM) in Figure 6B and Appendix A. Similar ATP-Glo results were found using alisertib, another AURKA inhibitor (Figure 6C and Appendix A). Our results demonstrated that the alisertib and TM combination dramatically sensitized EAC cells to the treatment. The IC50s of alisertib was decreased from 59.36 to 27.65 nM in FLO-1 and from 64.96 to 27.15 nM in OE33 cells. Meanwhile, the IC50s of TM was decreased from 185.9 to 16.59 ng/mL in FLO-1 and from 53.68 to 16.29 ng/mL in OE33 (Figure 6C and Appendix A).

The clonogenic cell survival assay was performed to test the long-term effect of AURKA inhibition during ER stress in FLO-1, OE33 cells with TM alone, alisertib alone, or combination treatment. The results indicated that alisertib and TM combination treatment significantly decreased cell viability more than a single treatment, quantified by relative cell staining intensity in a long-term colony formation assay (Figure 6D and Appendix A). To investigate whether there is a synergistic effect of AURKA inhibition and IRE1α inhibition, we tested the IRE1α inhibitor 3′6′-DMAD in FLO-1, OE33, and SK-GT4 cells. The ATP-Glo data indicated similar IC50s for all three cell lines from 153.7 to 175.52 nM (Appendix A). However, there was no synergistic effect of alisertib and 3′6′-DMAD in all three cell lines (Appendix A; OE33 data are not shown). In conclusion, these novel results indicated that AURKA inhibition combined with TM-induced ER stress could be a possible therapeutic approach for EAC.

### 3.7. AURKA and ABS Promote IRE1α Phosphorylation In Vivo

Tissues from FLO-1 and OE33 tumor xenograft mouse models [29] were investigated to examine the regulation of IRE1α by AURKA in vivo. Immunofluorescence staining of the xenograft tumor tissue samples demonstrated that alisertib treatment not only decreased xenograft tumor size but also significantly reduced phospho-IRE1α (p-IRE1α) staining, as compared with control tumor samples (Figure 7A,B and Appendix A, *p* < 0.01). Immunofluorescence staining data also demonstrated colocalization of AURKA and IRE1α (yellow staining), higher in the control group than in the alisertib treatment group (Figure 7A and Appendix A). To further confirm the IRE1α activation after bile salts in vivo, we used the pL2-IL1β mouse, and a model of chronic esophageal inflammation that develops BE-like lesions was investigated [30]. Immunofluorescence staining was performed for p-IRE1α and AURKA in normal control esophagus and gastroesophageal junction high-grade dysplasia (HGD) samples. p-IRE1α staining was remarkably stronger in HGD samples compared with normal tissues. Furthermore, we observed p-IRE1α and AURKA colocalization in HGD samples.

## 4. Discussion

Esophageal adenocarcinoma is an aggressive malignancy with an estimated 16,000 deaths in the United States in 2020 [41]. Chronic GERD is considered the leading risk factor for developing a premalignant metaplastic condition known as BE and its progression to EAC [42]. However, the current understanding of the underlying biology and molecular mechanisms of EAC remains limited. Excessive protein production places cancer cells in endless ER stress. Therefore, cancer cells must adapt and tilt the balance of survival over death during ER stress by activating unfolded protein response (UPR) [43]. In this study, we demonstrated the role of AURKA in promoting prosurvival UPR under reflux conditions in EAC.

Although the role of AURKA in promoting cancer cell survival and drug resistance has been established [14,44,45], its role in regulating prosurvival UPR in cancer cells remains unknown. Analysis of TCGA, GEO databases, and human tissue samples demonstrated a robust positive correlation between AURKA and increased UPR/ER stress in EAC. Recent studies indicated that IRE1α, PERK, and ATF6 are essential ER stress sensors, regulating the balance between cell survival and death during ER stress through UPR [33]. Our data not only indicated a remarkable increase in UPR proteins in EAC cells but also showed the critical prosurvival role of IRE1α compared with the other two primary ER stress sensors, PERK and ATF6. As we showed the co-overexpression of AURKA and UPR key proteins, our data indicated that AURKA induced IRE1α levels and phosphorylation in EAC cells. IRE1α and its downstream sXBP1 promote cell survival during ER stress in several malignancies [33,46].

EAC is a unique malignancy that develops under continuous cellular stress mediated by chronic GERD, forcing cells to develop adaptive prosurvival cellular mechanisms to overcome reflux-induced stress [47]. Surprisingly, to the best of our knowledge, no studies in the literature have investigated the UPR pathway under reflux/ABS conditions in EAC. Our results indicate that AURKA-induced IRE1α tilts the cellular homeostatic balance towards cell survival rather than death during an overwhelming ER stress under oxidative genotoxic reflux conditions. Of note, we detected colocalization and binding of AURKA to IRE1α during reflux/ABS conditions in vitro and in vivo. At the same time, neither ABS treatment nor AURKA affected BIP binding to IRE1α. Based on our data, we suggest that AURKA binding to IRE1α enhances IRE1α stability and autophosphorylation under ER stress conditions without affecting BIP binding to IRE1α. Autophosphorylation of IRE1α has been described as a critical mechanism for counteracting cell death and promoting survival under ER stress [40,48]. Our findings elucidate a previously unrecognized mechanism by which AURKA activates UPR in EAC cells to promote cell survival under reflux conditions.

To examine the efficacy of targeting AURKA and ER stress in EAC, we combined AURKA inhibition and ER stress inducer treatment. Our data indicated that both AURKA knockdown and pharmacological inhibition synergize with an ER stress inducer in short-or long-term experimental settings. Similar findings were discovered in different cancers where TM synergized with several antitumor therapeutics [49,50,51]. In the meantime, we also tested the efficacy of AURKA inhibition and IRE1α inhibition in EAC. The results demonstrated a lack of substantial synergistic effect when AURKA inhibition is combined with IRE1α inhibition. These findings are expected and provide additional proof that AURKA inhibition is sufficient to inhibit IRE1α, not necessitating the addition of IRE1 inhibitors. Future therapeutic strategies should focus on treating EAC cells with the combination of an AURKA inhibitor and ER stress inducer.

## 5. Conclusions

In summary, our findings demonstrate for the first time the function of AURKA in regulating prosurvival UPR through IRE1α. This AURKA-dependent UPR activation provides a new paradigm in esophageal tumorigenesis and treatment.

## Figures and Tables

**Figure 1 cancers-14-01401-f001:**
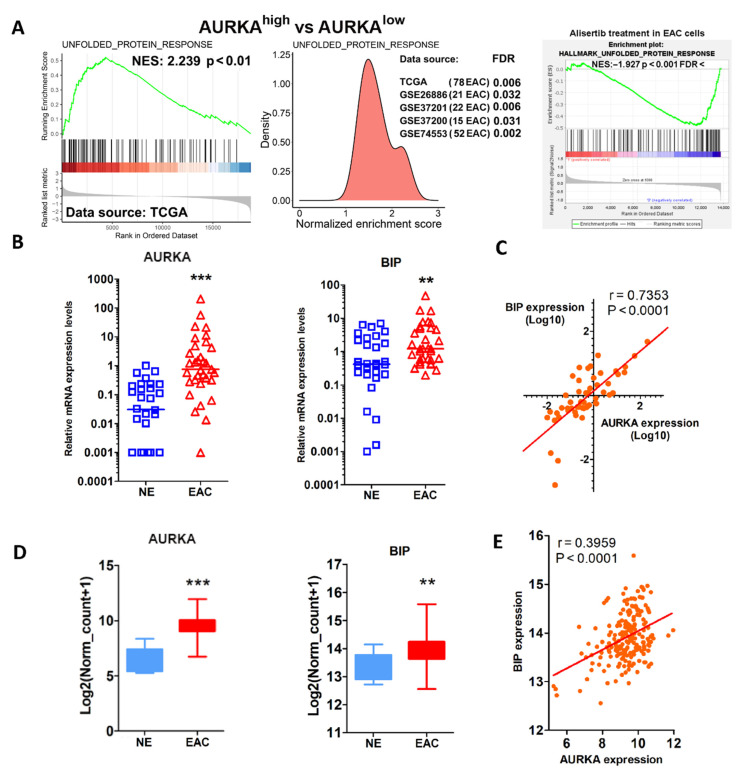
AURKA is positively correlated with UPR in EAC. (**A**) Left and middle panels: TCGA and GEO database analysis of signal pathway enrichment in AURKA high EAC compared with AURKA low EAC. Right panel: unfolded protein response gene set enrichment analysis in FLO-1, OE33, and SK-GT4 cell NGS data of alisertib treatment groups compared with control groups. (**B**) qRT-PCR analysis of AURKA and BIP mRNA expression in the normal human esophagus (NE) and esophageal adenocarcinoma (EAC) tissue samples. (**C**) Correlation between AURKA and BIP mRNA expression in esophageal tissue samples from (**B**). (**D**) Data analysis of AURKA and BIP mRNA expression in the normal esophagus (NE) and esophageal adenocarcinoma (EAC) samples from TCGA database. (**E**) Correlation between AURKA and BIP mRNA expression in esophageal tissue samples from TCGA database. ** *p* < 0.01, *** *p* < 0.001.

**Figure 2 cancers-14-01401-f002:**
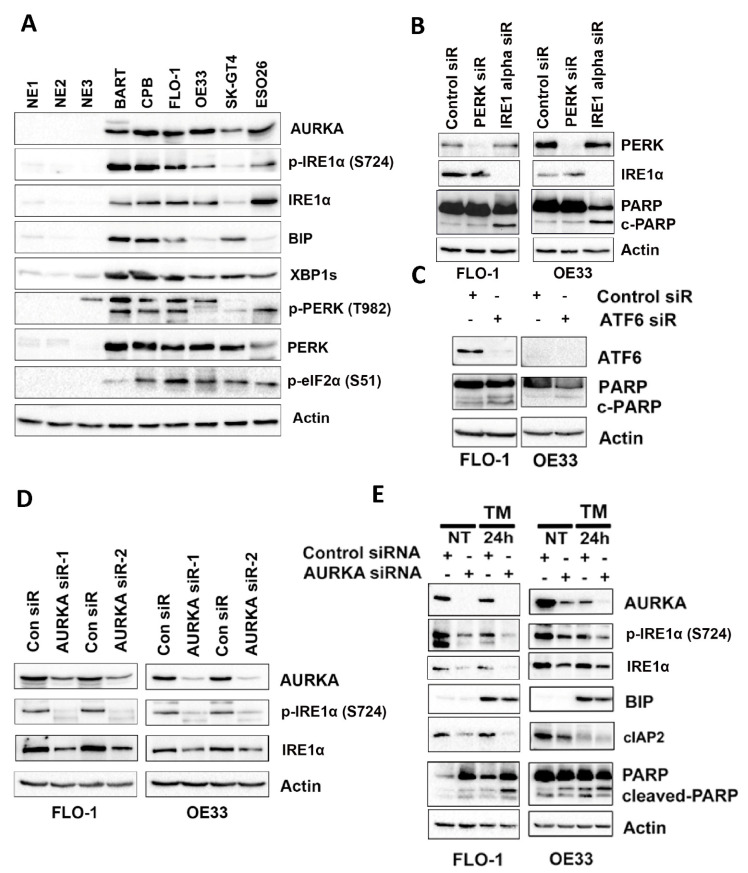
IRE1α promotes cancer cell survival dependent on AURKA in EAC. (**A**) Western blot analysis of AURKA, p-IRE1α (S724), IRE1α, BIP, XBP1s, p-PERK (T982), PERK, p-eIF2α (S51), and β-actin protein expression in the normal esophagus (NE), BART, CPB, FLO-1, OE33, SK-GT4, and ESO26 cells. (**B**) Western blot analysis of PERK, IRE1α, PARP, cleaved-PARP (c-PARP), and β-actin protein expression in FLO-1, OE33 cells at 72 h after control siRNA, PERK siRNA, or IRE1α siRNA transfection. (**C**) Western blot analysis of ATF6, PARP, cleaved-PARP (c-PARP), and β-actin protein expression in FLO-1 and OE33 cells at 72 h after control siRNA or AFT6 siRNA transfection. (**D**) Western blot analysis of AURKA, p-IRE1α (S724) IRE1α, and β-actin protein expression in FLO-1 and OE33 cells at 72 h after control siRNA, AURKA siRNA-1, or AURKA siRNA-2 transfection. (**E**) Western blot analysis of AURKA, p-IRE1α (S724), IRE1α, cIAP2, BIP, PARP, cleaved-PARP (c-PARP), and β-actin protein expression in FLO-1 and OE33 cells at 72 h after control or AURKA siRNA transfection with or without tunicamycin (TM) 10 µg/mL treatment for the last 24 h after transfection.

**Figure 3 cancers-14-01401-f003:**
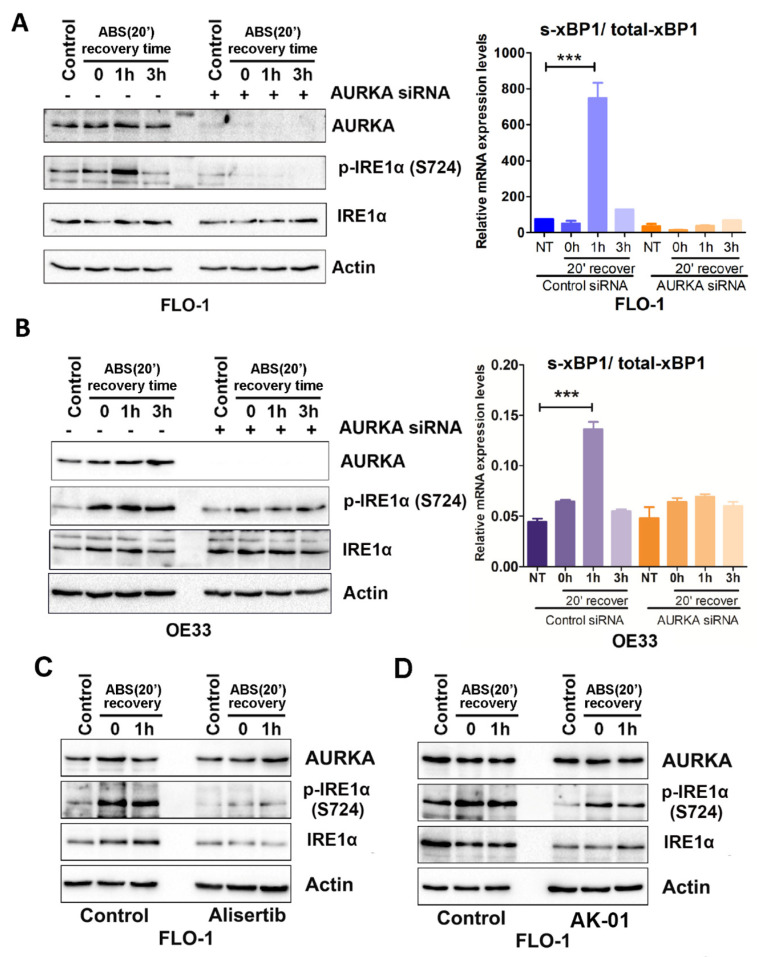
ABS induces IRE1α phosphorylation and activation depending on AURKA in EAC (**A**). Left panel: Western blot analysis of AURKA, p-IRE1α (S724), IRE1α, and β-actin protein expression in control siRNA or AURKA siRNA transfected FLO-1 cells with control or ABS (pH 4.0, 200 µM, 20′) exposure followed by recovery at indicated time courses (0, 1 h or 3 h). Right panel: qRT-PCR analysis of the ratio of spliced-xBP1 (s-xBP1)/total-xBP1 in the same cells as (**A**) left panel. (**B**) Similar results in OE33 cells as in (**A**). (**C**) Western blot analysis of AURKA, p-IRE1α (S724), IRE1α, and β-actin protein expression in FLO-1 control cells and alisertib 200 nM 48 h pretreated cells with control or ABS (pH 4.0, 200 µM, 20′) exposure followed by recovery at indicated time courses (0 or 1 h). (**D**) Similar results as in C in FLO-1 control cells and cells with AK-01 200 nM 48 h pretreatment. *** *p* < 0.001.

**Figure 4 cancers-14-01401-f004:**
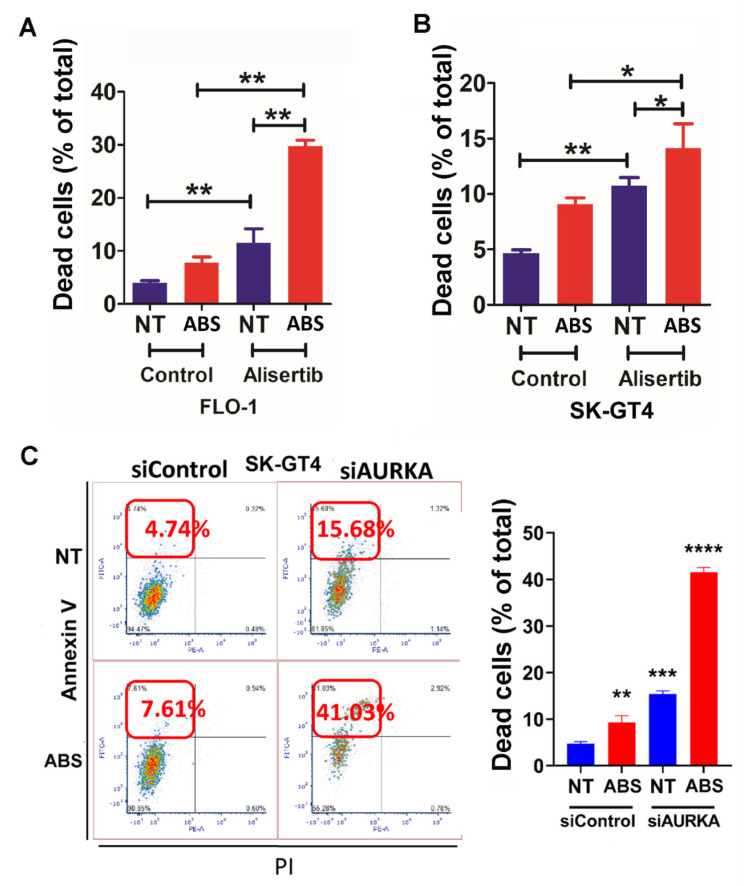
AURKA protects EAC cells from ABS-induced cell death. (**A**) Apoptotic cell quantification based on flow cytometry analysis of Annexin V and PI staining in FLO-1 cells with alisertib (200 nM, 24 h), ABS (pH 4.0, 200 µM, 20′ + 3 h recovery), or a combination. (**B**) Similar results in SK-GT4 cells as in (**A**). (**C**) Left panel: flow cytometry analysis of Annexin V and PI staining in SK-GT4 cells treated with ABS (pH 4.0, 200 µM, 20′ + 3 h recovery) at 72 h after the transfection of control siRNA or AURKA siRNA. Right panel: quantification of Annexin V-positive and PI-negative staining cells from the left panel. * *p* < 0.05, ** *p* < 0.01, *** *p* < 0.001, **** *p* < 0.0001.

**Figure 5 cancers-14-01401-f005:**
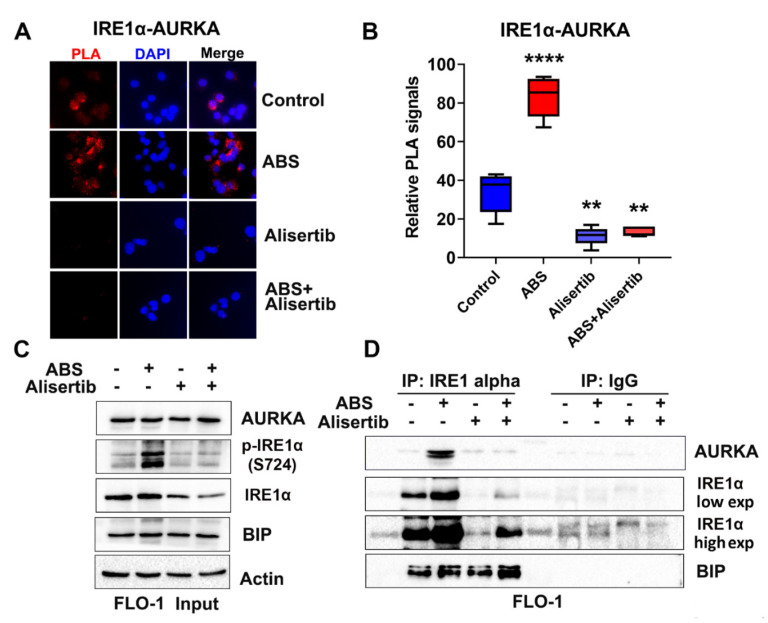
AURKA binds to IRE1α in ABS exposure. (**A**) In situ proximity ligation assay (PLA) immunofluorescence staining in FLO-1 control cells or alisertib 200 nM 48 h pretreated cells with or without ABS (pH 4.0, 200 µM, 20′) exposure. The red signal indicates the positive PLA signal. 200x magnification. (**B**) Quantification of positive PLA signal (red signal) in (**A**) using ImageJ software. (**C**) Western blot analysis of AURKA, p-IRE1α (S724), IRE1α, BIP, and β-actin protein expression in FLO-1 AURKA IP input samples. Cells were treated with alisertib 200 nM 48 h, followed by ABS (pH 4.0, 200 µM, 20′) exposure. (**D**) Western blot analysis of AURKA, IRE1α, and BIP protein expression in AURKA IP or IgG IP samples from C. ** *p* < 0.01, **** *p* < 0.0001.

**Figure 6 cancers-14-01401-f006:**
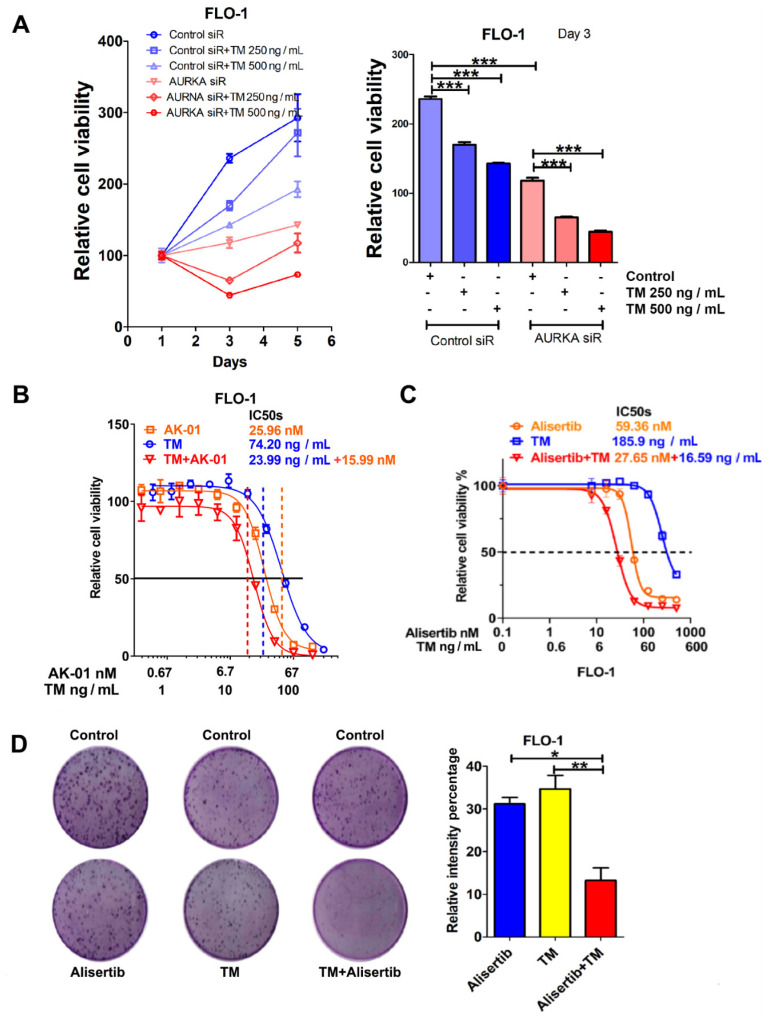
AURKA promotes cancer cell survival during tunicamycin-induced ER stress. (**A**) Left panel, ATP-Glo analysis of FLO-1 cells treated with control or tunicamycin (250 or 500 ng/mL for 3 or 5 days), combined with control siRNA or AURKA siRNA transfection. Right panel quantifies cell viability on day 3 of cells in (**A**). (**B**) ATP-Glo analysis of FLO-1 cells treated with tunicamycin, AURKA inhibitor AK-01, or a combination for 5 days. (**C**) Similar results as the left panel in FLO-1 cells treated with tunicamycin, AURKA inhibitor alisertib, or a combination for 5 days. (**D**) Left panel, representative clonogenic cell survival assay wells in FLO-1-treated tunicamycin, AURKA inhibitor alisertib, or a combination. Right panels, quantification of the left panel. * *p* < 0.05, ** *p* < 0.01, *** *p* < 0.001.

**Figure 7 cancers-14-01401-f007:**
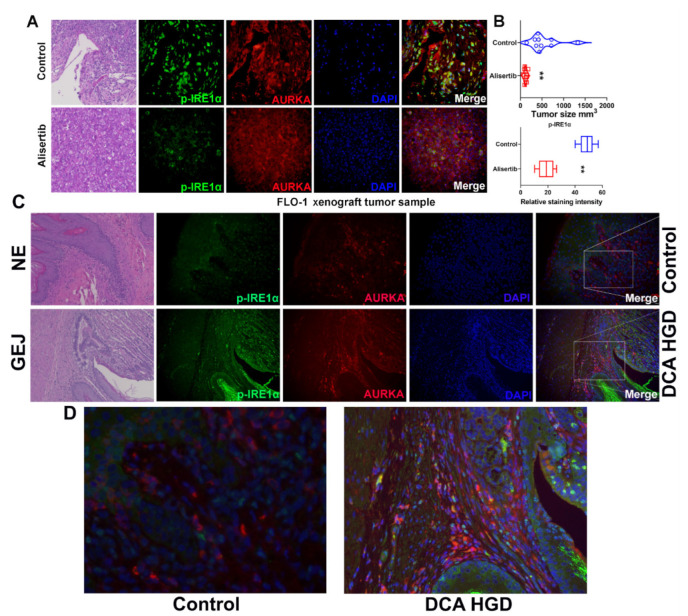
AURKA and ABS induce IRE1α phosphorylation in vivo. (**A**) H&E staining (left panels) and IF staining of p-IRE1α (S724, green signal), AURKA (red signal) in FLO-1 xenograft tumor samples with or without alisertib treatment. 200× magnification. (**B**) Upper panel: tumor size of FLO-1 xenograft tumors, 200× magnification. Lower panel: quantification of p-IRE1α staining in (**A**) using ImageJ. (**C**) H&E staining (left panel) and IF staining of p-IRE1α (S724, green signal), AURKA (red signal) in gastroesophageal junction tissues from the normal esophagus (NE) of wide-type mouse and high-grade dysplasia (HGD) of deoxycholic acid (DCA)-fed L2-IL1β transgenic mouse, 200× magnification. ** *p* < 0.01. (**D**) Partially enlarged picture of (**C**) 600× magnification.

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
