# Peer review of "Unfolded Protein Response Is Activated by Aurora Kinase A in Esophageal Adenocarcinoma"

_cancers, 2022, doi:10.3390/cancers14061401_

Round 1

Reviewer 1 Report

This is an interesting study presented by Lu et al. that demonstrated a novel AURKA-IRE1α interaction and the role of the AURKA-IRE1α axis in regulating unfolded protein response in esophageal adenocarcinoma. The co-existence, physical interaction, physiological causality, and clinical relevance of the AURKA-IRE1α axis were clearly analyzed and reported in this manuscript. Hope my comments below are helpful to the authors:

  1. AURKA is highly relevant with cell cycle progression. Are any of the presented AURKA expression variances in patient samples caused by their differential proliferative capacity?
  2. Alisertib causes G2/M arrest that treatment with such an inhibitor for 48 hours may result in reduced cell density and modulated cell growth and metabolic states, compared to the untreated cells. These factors feedback indirectly to the UPR pathways. Could the experiment in Fig. 5 be done with a more transient AURKA inhibition? 
  3. The authors showed that AURKA-induced IRE1α phosphorylation is a response to ABS stimulation. Is AURKA itself phosphorylated upon ABS treatment? Which phosphorylation site(s) on AURKA controls the activity required for downstream IRE1α phosphorylation?
  4. Please correct typos and errors.

Author Response

Please see our response in the attached response letter. Thanks!

Reviewer 2 Report

  1. This manuscript is interesting and well-done.
  2. The strength of this article is well organized for readers to understand for issue of cancer therapy in EAC.
  3. This research proveded that AURKA promotes survival under tunicamycin induced ER stress in EAC cells.
  4. Theme of this research, to suggest a better therapeutic approach of cancer therapy through AURKA under treatment of tunicamycine. Moreover revealed to IRE1α > AURKA > EAC Cell (refractory cancer) Survival. However, all authors should be cautious about “results were easy to anticipate”.
  5. I found this paper interesting, only few comments are addressed: Future and prospective treatment strategies should be briefly mentioned based on the authors' experience.
  6. Line 276, is there a reason why you chose cleaved PARP as a apoptosis marker?
  7. Line 278, if down modulated PERK or ATF6, how is expression of CHOP or Bcl-2? There's no need to modify the current figures.
  8. What do you think about relationshhip between drug resistant cancer and AURKA?

Author Response

(The authors gave the same response as above.)
